# Dual function of Zika virus NS2B-NS3 protease

**Sergey A. Shiryaev**[1]*, **Piotr Cieplak**[1], **Anton Cheltsov**[2], **Robert C. Liddington**[1], **Alexey V. Terskikh**[1]*

1 Sanford-Burnham-Prebys Medical Discovery Institute, Infectious and Inflammatory Disease Center, La Jolla, California, United States of America, 2 Q-mol LLC, San Diego, California, United States of America

* sshiryaev2000@gmail.com (SAS); terskikh@sbp.edu (AVT)

## Abstract

Zika virus (ZIKV) serine protease, indispensable for viral polyprotein processing and replication, is composed of the membrane-anchored NS2B polypeptide and the N-terminal domain of the NS3 polypeptide (NS3pro). The C-terminal domain of the NS3 polypeptide (NS3hel) is necessary for helicase activity and contains an ATP-binding site. We discovered that ZIKV NS2B-NS3pro binds single-stranded RNA with a $K_d$ of ~0.3 µM, suggesting a novel function. We tested various structural modifications of NS2B-NS3pro and observed that constructs stabilized in the recently discovered "super-open" conformation do not bind RNA. Likewise, stabilizing NS2B-NS3pro in the "closed" (proteolytically active) conformation using substrate inhibitors abolished RNA binding. We posit that RNA binding occurs when ZIKV NS2B-NS3pro adopts the "open" conformation, which we modeled using highly homologous dengue NS2B-NS3pro crystallized in the open conformation. We identified two positively charged fork-like structures present only in the open conformation of NS3pro. These forks are conserved across *Flaviviridae* family and could be aligned with the positively charged grove on NS3hel, providing a contiguous binding surface for the negative RNA strand exiting helicase. We propose a "reverse inchworm" model for a tightly intertwined NS2B-NS3 helicase-protease machinery, which suggests that NS2B-NS3pro cycles between open and super-open conformations to bind and release RNA enabling long-range NS3hel processivity. The transition to the closed conformation, likely induced by the substrate, enables the classical protease activity of NS2B-NS3pro.

## Author summary

Zika virus (ZIKV) serine protease is indispensable for viral replication and is composed of the protease and helicase subunits. We discovered that the ZIKV protease subunit binds RNA, suggesting a novel function. We tested various structural modifications and conformations of ZIKV protease and observed that only one of three possible conformations, the "open" proteolytically inactive conformation, binds RNA. We identified specific 3-dimensional structures present only in the open conformation of many flaviviruses, including ZIKV and Dengue, suggesting that our finding may have a broad implication for several flaviviruses. We propose a novel "reverse inchworm" model for helicase-protease machinery that is key for viral propagation and possible novel ways of targeting flaviviruses.

**Data Availability Statement:** All relevant data are in the manuscript and its supporting information files.

**Funding:** National Institute of Neurological Disorders and Stroke (NINDS): R01 NS105969 to

A.V.T. A.V.T. and S.A.S. received salary from
NINDS. The funders had no role in study design,
data collection and analysis, decision to publish, or
preparation of the manuscript.

**Competing interests:** The authors have declared
that no competing interests exist.

## Introduction

Zika virus (ZIKV) is a member of the *Flaviviridae* family that includes West Nile virus
(WNV), dengue virus (DENV serotypes 1–4), Japanese encephalitis virus, hepatitis C virus
(HCV), and tick-borne encephalitis virus) among many other human pathogens. Like other
flaviviruses with recurrent outbreaks, ZIKV is considered a major global health threat [1,2].
ZIKV infection of pregnant mothers can cause microcephaly in the fetus [3–5] and infection
of adults has been linked to the autoimmune neurodegenerative disease Guillain–Barré syn-
drome [6–8]. Unfortunately, to date, no approved medications or vaccines exist to treat or pre-
vent ZIKV infection in the United States [9–11].

ZIKV is a positive-sense, single-stranded RNA virus, which is similar to other flaviviruses.
Its genome is ~11 kb and encodes a 3423-amino acid polyprotein precursor, which is com-
posed of 3 structural and 7 nonstructural proteins. Once inserted into the host endoplasmic
reticulum (ER) membrane, the viral polyprotein is cleaved by a viral protease at cytosol-
exposed junctions between nonstructural proteins and within the capsid protein C. Host pro-
teases, such as furin and signal peptidase, process the polyprotein within the ER (**S1A Fig**).

ZIKV NS3 protein is an ~615 aa multifunctional protein exhibiting endopeptidase, RNA
helicase, RNA triphosphatase, and NTPase activities [12]. The N-terminal 170-aa domain of
NS3 (NS3pro) interacts with a cytosolic 48-aa region of the membrane-anchored NS2B to
form a highly active serine protease termed NS2B-NS3pro (**S1B Fig**) [13]. The critical depen-
dance of viral propagation on NS2B-NS3pro has made this polypeptide a promising antiviral
drug target. However, the development of competitive inhibitors is challenging because of the
high structural homology between the active centers of viral NS2B-NS3pro and multiple cellu-
lar serine proteases with critical cellular functions [14]. The development of allosteric anti-
NS2B-NS3pro inhibitors can overcome this obstacle [14,15].

Recently, we determined the crystal structure of ZIKV NS3pro with the covalently linked
NS2B cofactor with 2.5 Å and 3 Å resolution. These structures revealed that NS2B-NS3pro
adopts a "super-open" conformation (PDB IDs: 5TFN, 5TFO, 6UM3, and 7M1V) that is quite
distinct from the previously reported proteolytically active "closed" conformation of ZIKV
NS2B-NS3pro (PDB ID: 5LC0, 16]. Although the crystal structure of ZIKV protease in the
"open" conformation has not been reported, the structures of DENV and WNV NS2B-NS3pro
in the open conformation (PDB IDs: 2FOM and 2GGV) suggest that ZIKV protease open con-
formation can be modeled using closely related DENV and WNV.

The NS3-NS4A protease domain of HCV is distantly homologous to ZIKV NS2B-NS3pro;
however, the two differ in terms of the single conformation of HCV protease domain, distinct
substrate specificity, and positive charge in the active site of the former [17]. Several indepen-
dent studies have demonstrated that the protease domain of HCV NS3 binds to RNA, and full-
length HCV NS3 protein possesses better dsRNA processivity than the helicase domain alone
[18–20]. However, the isoelectric point (pI) of ZIKV NS2B-NS3pro is lower than that of HCV
protease, which implies that it may not be able to bind RNA. Nevertheless, the full-length NS3
(NS3pro+NS3hel) of the homologous WNV and DENV2 viruses also demonstrate signifi-
cantly better processivity compared with the corresponding NS3 helicase domains alone
[21,22].

Here for the first time, we demonstrated that the ZIKV NS2B-NS3pro protein binds RNA
with a physiologically meaningful affinity. Further examination of different conformational
states suggested that RNA binds to the open conformation of ZIKV NS2B-NS3pro. We dem-
onstrated that RNA binding interfered with the proteolytic activity of NS2B-NS3pro[23] and
identified small molecule inhibitors that prevent RNA binding to NS2B-NS3pro. Molecular
modeling identified conserved fork-like structures able to accommodate RNA. Modeling of

NS2B-NS3pro-NS3hel revealed a positively charged continuous interface between the protease and the helicase. Based on our results, we propose a novel "reverse inchworm" mechanism of coupled RNA and polypeptide processing by the ZIKV NS2B-NS3 machinery, which is likely to be a shared feature of all flaviviruses.

## Results

### Super-open conformation is conserved between ZIKV and JEV

In addition to our structure of ZIKV NS2B-NS3pro (PDB ID 7M1V), the crystal structure of another orthologous flavivirus, Japanese encephalitis virus (JEV) NS2B-NS3pro (PDB ID 4R8T) was uncovered in 2015[24]. However, no structural or functional analysis was conducted for PDB ID 4R8T crystal structure, likely explaining the lack of attention paid by the research community. We computed the overlay of the ZIKV NS2B-NS3 protease structures in the super-open confirmation of ZIKV NS2B-NS3pro (PDB ID 7M1V) with the crystal structure of JEV NS2B-NS3pro (PDB ID 7M1V) (**S2 Fig**). We observed an almost identical organization of the critical NS3pro C-terminal loop between these two structures (RMSD 0.6A).

### Modeling of ZIKV NS2B-NS3pro open conformation

We took advantage of the structures DENV and WNV NS2B-NS3pro crystalized in the open conformation (PDB IDs: 2FOM and 2GGV) to model ZIKV NS2B-NS3pro with the help of FFAS [25], MODELLER [26], and RoseTTAFold [27] packages. Additional insights were obtained by modeling the withdrawal of the C-terminal part of NS2B from a hydrophobic cleft of NS2B-NS3pro in the closed conformation [23]. When compared to the open conformation, the super-open conformation induces further dissociation of NS2B from NS3pro and a refolding of NS3pro C-terminal residues, which dramatically affects the overall fold of NS3pro. Given the apparent continuum of these structural changes, we propose that NS2B-NS3pro can dynamically switch between closed, open, and super-open conformations (**Figs 1 and S3**).

### ZIKV NS2B-NS3pro polypeptide binds RNA

To investigate the binding of RNA and single-stranded (ss) DNA to the ZIKV NS2B-NS3pro polypeptide, we employed a classical Fluorescence Polarization (FP) assay. Fluorescein amidites (FAM)-labeled RNA or ssDNA were mixed with NS2B-NS3pro and an increase in FP was used to detect binding [28]. The wild-type proteolytically competent NS2B-NS3pro polypeptide, which can adopt closed, open, or super-open conformations, demonstrated a strong affinity for RNA. The direct binding assays and competition assays yielded similar results, both indicating an equilibrium dissociation constant ($K_d$) of ~0.34 μM (**Fig 2A and 2B**). An increase in FP signal was also obtained upon incubation of NS2B-NS3pro with FAM-labeled ssDNA (**Fig 2C**). However, both the direct binding and competition assays yielded $K_d$ values of ~2.0 μM for ssDNA that was ~10-fold lower than that for RNA (**Fig 2C and 2D**). The unlabeled RNA and ssDNA efficiently competed with the FAM-labeled probes with similar kinetics (**Fig 2B and 2D**), confirming the specificity of binding.

To investigate RNA binding to ZIKV NS2B-NS3pro polypeptide in more detail, we generated a set of additional constructs aimed at probing some key structural and conformational requirements (**S4 Fig**). First, we tested the property of a negatively charged 12-aa linker region ([171]EEETPVECFEPS[182]) connecting the NS3pro domain with the NS3 helicase domain. The NS2B-NS3-long construct (which contains 182 N-terminal residues of NS3) was crystallized in the super-open conformation (PDB ID 5TFN). Given the unstructured nature of the linker (which is naturally present within the NS3 polypeptide) the NS2B-NS3-long construct should

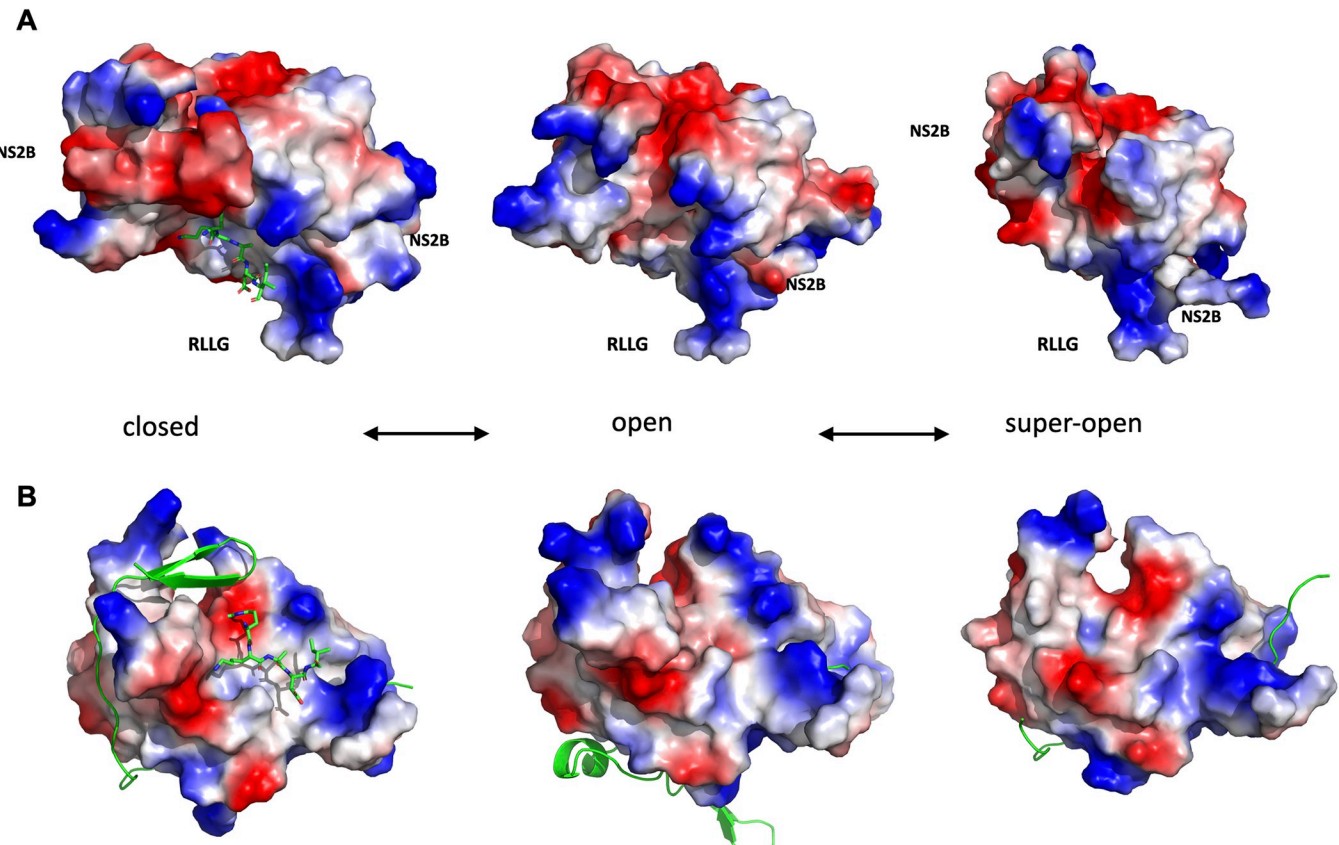

**Fig 1. The conformational landscape of ZIKV NS2B-NS3pro. (A)** Transitional equilibrium between closed, open, and super-open conformations of ZIKV NS2B-NS3pro. A peptide-based substrate (RKADI, green ball and stick model) in the protease active center is modeled on the related structure of WNV NS2B-NS3pro + aprotinin (PDB ID: 2IJO). The RLLG loop faces the surface of the ER membrane. **(B)** Same conformations as in B after rotating up by 90 degrees along the horizontal axis. The active site is now facing the viewer. NS2B is shown in green cartoon structure. Positively charged side chains are blue colored, and negatively charged residues are red.

be capable of adopting all conformations. Surprisingly, incubation of NS2B-NS3-long with RNA or ssDNA did not increase the FP signal, indicating a lack of RNA or ssDNA binding (**Fig 3A**). This observation may indicate that the 12-aa negatively charged linker region modulates RNA binding to NS2B-NS3pro, potentially competing with negatively charged RNA. The exact role of this linker, which is bound at both termini in the full-length NS3, is incompletely understood.

Next, we tested several mutant constructs designed to stabilize the super-open conformation. The truncated NS2B-NS3-short construct lacked amino acids 161–170, thus unable to assume either the closed or open conformations (PDB 5LC0, 23]. FP assays failed to detect RNA or ssDNA binding to such NS2B-NS3-short mutant (**Fig 3A**), indicating a requirement for aa 161–170 of NS3pro to support RNA binding and suggesting that super-open conformation is refractory to RNA binding.

We next tested the NS2B-NS3pro-Mut5 and Mut7 constructs, which were designed to be stabilized in the super-open conformation *via* a disulfide bond between two cysteines introduced into the NS3 sequence. We solved the structures of Mut5 and Mut7 and confirmed that both mutants adopt the super-open conformation (PDB IDs 6UM3, 7M1V). We demonstrated that neither of these mutants was able to bind RNA (**Fig 3A**), suggesting that the super-open conformation is refractory to RNA binding.

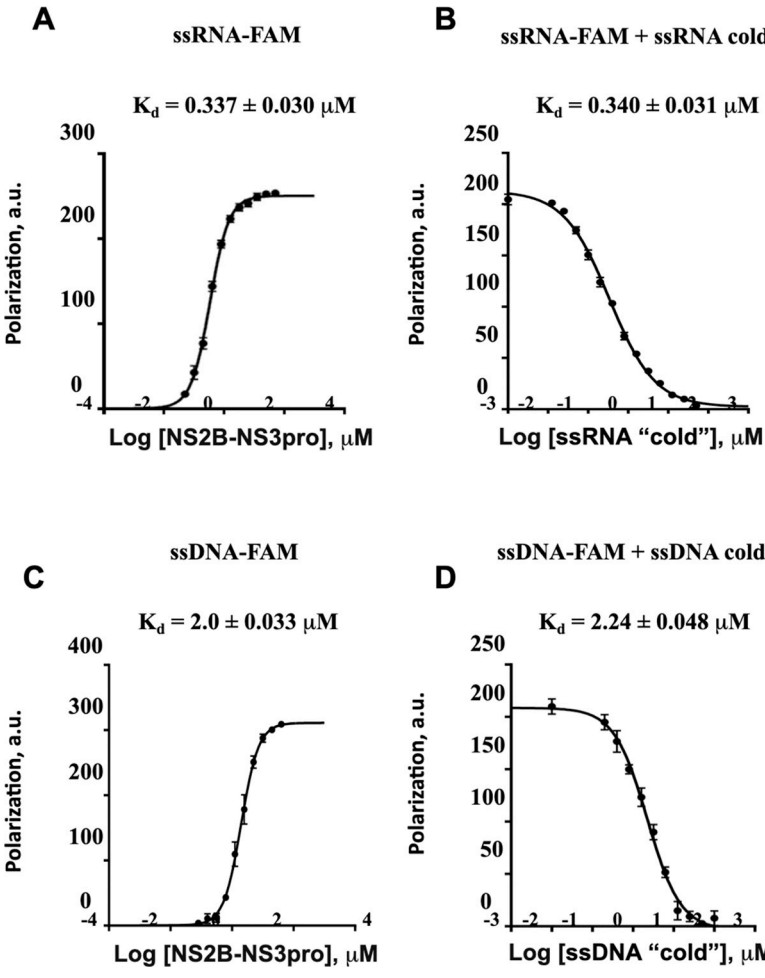

**Fig 2. ZIKV NS2B-NS3pro binds ssRNA and ssDNA.** Fluorescent polarization assays of **(A)** Fluorescein amidites (FAM)-labeled ssRNA (20 poly-rU) binding to the indicated amounts of NS2B-NS3pro, **(B)** FAM-labeled ssRNA (20 poly-rU) binding to 1 μM of NS2B-NS3pro in the presence of the indicated amounts of unlabeled ssRNA (20 poly-rU). **(C, D)** As for A, B, except experiments were performed with FAM-labeled ssDNA (20 poly-dT).

Taken together, these experiments suggest that while only wild type NS2B-NS3pro polypeptide efficiently binds RNA while all structural modifications tested so far, particularly those that lock NS2B-NS3pro in the super-open conformation, abrogate RNA binding. Further, the protease-helicase linker region (featuring 5 negative charges) appears to impede RNA binding to the wild-type construct possibly by competing with the RNA.

## NS2B-NS3pro substrate-mimicking inhibitors compete with RNA binding

Several groups have demonstrated that protease activity inhibitors (peptidomimetics or small molecule compounds) that latch onto NS2B-NS3pro active center stimulate protease transition onto a closed conformation [16,23,29–31]. This phenomenon has been well documented for proteases from several *Flaviviridae*, including WNV (PDB IDs 2IJO, 2YOL) [23,29], DENV-2 (PDB ID 3U1J) [30], and ZIKV (PDB IDs 5YOF, 5LCO) [16,31]. To determine whether RNA can bind to ZIKV NS2B-NS3pro in the closed, proteolytically active state, we induced this conformation in NS2B-NS3pro using two substrate-mimicking inhibitors; aprotinin, a 70-aa serine protease inhibitor, and WRPK3, a synthetic peptide inhibitor that covalently binds to the NS2B-NS3pro active center [32].

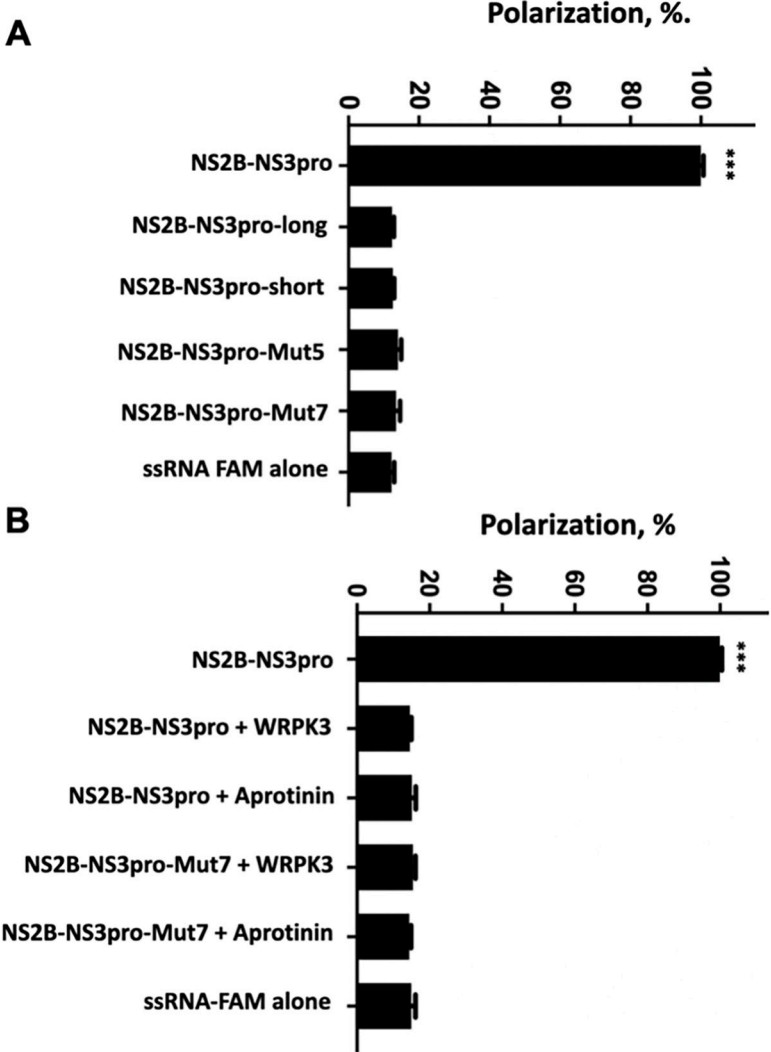

**Fig 3. Structural and functional alterations in ZIKV NS2B-NS3pro preclude ssRNA binding. (A)** Modifications of NS2B-NS3pro (see S1 Fig for details) that prevent efficient binding of FAM-labeled ssRNA (20 poly-rU). **(B)** Inhibition of NS2B-NS3pro catalytic activity by aprotinin (~0.15 μM) and WRPK3 (covalent inhibitor) blocks ssRNA binding. Each condition is compared to protease NS2B-NS3pro, *** p<0.0005, two-tailed unpaired t-test with Welch's correction.

ZIKV NS2B-NS3pro was incubated with aprotinin or WRPK3 and the gel-filtration purified complexes were tested in the RNA FP assays. We observed no RNA binding to either of the complexes (**Fig 3B**). Since the active site in the closed conformation of ZIKV protease is negatively charged (**Fig 1A**), direct binding of RNA to the same site as aprotinin or WRPK3 seems unlikely. We posit that substrate-mimicking inhibitors eliminate RNA binding by inducing and sustaining the closed conformation of NS2B-NS3pro, which is incompatible with RNA binding. Such interpretation gains further credibility from our modeling results below.

## RNA binding inhibits the proteolytic activity of ZIKV NS2B-NS3pro

The experiments described thus far suggest that RNA doesn't bind to NS2B-NS3pro constructs in the closed or super-open conformations. By inference, this suggests that RNA binding

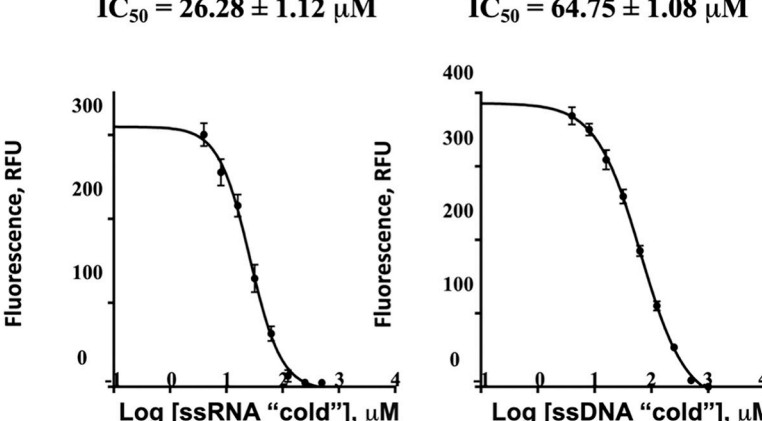

$IC_{50} = 26.28 \pm 1.12\ \mu M$        $IC_{50} = 64.75 \pm 1.08\ \mu M$

**Fig 4. ssRNA inhibits the proteolytic activity of NS2B-NS3pro.** Inhibition of ZIKV NS2B-NS3pro proteolytic activity by ssRNA (20-rA, left) and ssDNA (20-dA, right).

occurs when NS2B-NS3pro adapts an open proteolytically inactive conformation. We then investigated whether binding of RNA interferes with the proteolytic activity of NS2B-NS3pro. Indeed, using fluorogenic substrate cleavage assays, we found that binding of RNA or ssDNA inhibited the proteolytic activity of ZIKV NS2B-NS3 protease with $IC_{50}$ values of $26.28 \pm 1.12\ \mu M$ for RNA and $64.75 \pm 1.08\ \mu M$ for ssDNA (**Fig 4**). The observed inhibition of proteolytic activity is likely due to RNA-induced stabilization of NS2B-NS3pro, which adopts an open proteolytically inactive conformation [23,33].

## Allosteric inhibitors of NS2B-NS3 protease interfere with RNA binding

The open conformation of NS2B-NS3pro is achieved by the rearrangement of the NS2B cofactor (partial dissociation from NS3pro), leading to a loss of proteolytic activity [23,33]. NS2B dissociation uncovers a hydrophobic cleft in NS3pro, which presents a novel druggable site that is relatively conserved between different flaviviruses. Previously, we identified allosteric inhibitors of ZIKV, WNV, and DENV2 NS2B-NS3pro with sub- or low-micromolar $IC_{50}$ values *in vitro* [13,14]. These allosteric inhibitors were designed *in silico* to bind a region of NS2B-NS3pro distant from the proteolytic active site [14,34,35]. Despite a potent inhibition of several flaviviral proteases, we demonstrated that these inhibitors had no detectable effect on host serine proteases (furin and other proprotein convertases) that have a similar substrate specificity [13].

Here we investigated the effect of several ZIKV, WNV, and DENV2 NS2B-NS3pro inhibitors on RNA binding to ZIKV NS2B-NS3pro. We discovered that several of these compounds are also potent inhibitors of RNA binding to ZIKV NS2B-NS3pro (**Fig 5A**) [13]. For instance, a potent inhibitor of RNA binding, NSC86314, is also a sub-micromolar inhibitor of ZIKV, WNV, and DENV protease [13] (**Fig 5A and 5B**) and a potent inhibitor of WNV and DENV2 replicons in cell-based assays with $IC_{50}$ values of <50 μM [13,34,35]. See Materials and Methods for details.

We co-crystallized NSC86314 with the Mut7 ZIKV NS2B-NS3pro construct, which was designed to enforce the super-open conformation and to minimize protein dimerization observed with the previous construct Mut5 (PDB ID 6UM3). The Mut7 crystal structure was solved and refined to a resolution of 1.8 Å (PDB ID 7M1V). The structure revealed that NSC83614 binds to a hydrophobic pocket near the C-terminus of NS3pro (**Fig 5C**) and suggests that NSC83614 stabilizes the super-open conformation thus preventing the transition into the

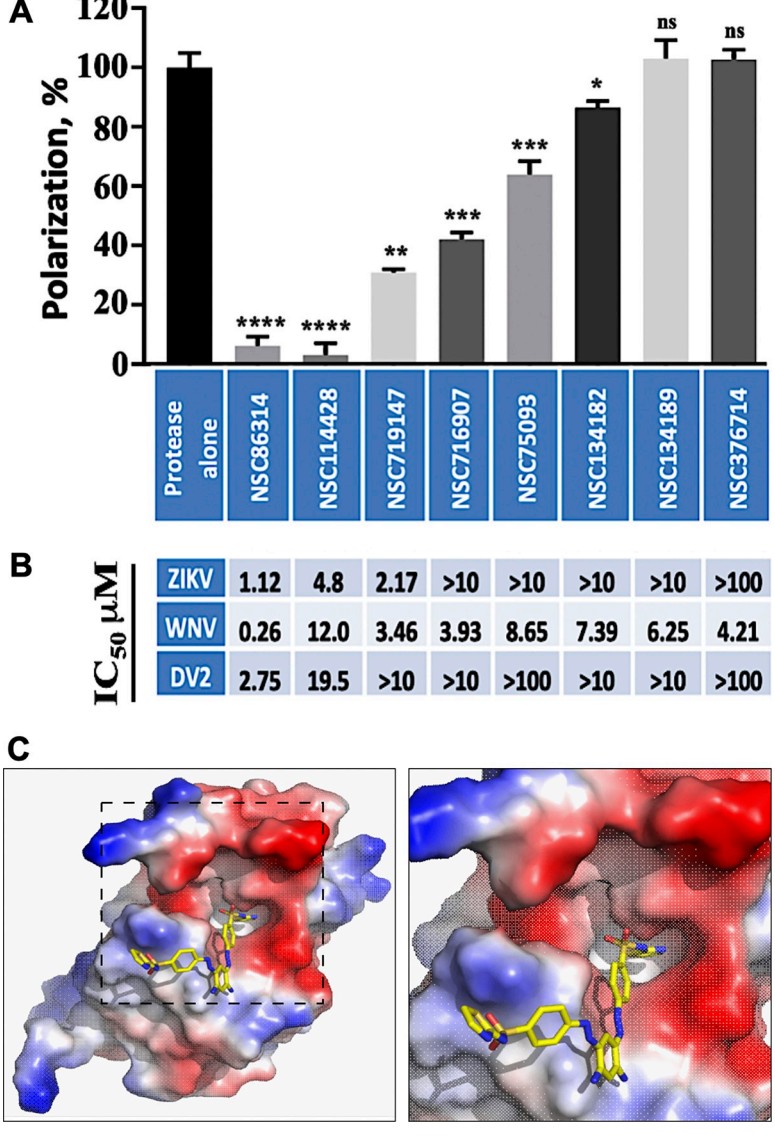

**Fig 5. Allosteric inhibitors of ZIKV NS2B-NS3pro interfere with RNA binding.** (A) Small allosteric inhibitors of ZIKV NS2B-NS3pro block ssRNA binding. Fluorescent polarization (%) using FAM labeled ssRNA (20 poly-rU). Each condition is compared to protease alone, * p<0.05, ** p<0.05, *** p<0.0005, **** p<0.0005 two-tailed unpaired t-test with Welch's correction. All inhibitors were assayed at 10 μM. (B) IC50 for inhibition of ZIKV, WNV, and DENV2 NS2B-NS3pro proteolysis by the corresponding inhibitors above (in A). (C) Crystal structure of ZIKV NS2B-NS3-Mut7 protease with the NSC86314 inhibitor (PDB 7M1V).

open conformation. However, because of the remaining dimerization of Mut7 additional predicted binding pocket was inaccessible for a part of NSC86314. Nevertheless, this crystal structure confirmed the feasibility of targeting novel pockets available in the super-open conformation of ZIKV NS2B-NS3pro.

We performed Virtual Ligand Screening (VLS) of the NCI Chemotherapeutic Agents Repository to find compounds having a similar scaffold structure to NSC86314. One of the compounds identified, NSC114428, was able to block both the proteolytic activity of ZIKV protease and RNA binding (**Fig 5A and 5B**) in the low μM range.

Next, we conducted a comparative analysis of flavivirus replication inhibitors with activity against ZIKV, WNV, and DENV proteases and their ability to inhibit RNA binding to ZIKV NS2B-NS3pro. The most potent inhibitors of ZIKV and DENV2 protease activity tended to also be the most potent inhibitors of RNA binding. However, this association was not observed for the inhibition of WNV protease activity, suggesting that the relationship between inhibition of flavivirus proteolytic activity and RNA binding may be complex (**Fig 5A and 5B**).

Taken together, these results provide a proof-of-principle for a novel class of allosteric inhibitors that specifically target newly identified druggable pockets present in the open and super-open conformations of ZIKV NS2B-NS3pro. Our results suggest that such allosteric inhibitors can interfere with the RNA-binding activities of NS2B-NS3pro in addition to blocking the protease activity.

## Modeling RNA binding to ZIKV NS2B-NS3

Our biochemical findings provided evidence that NS2-NS3pro binds RNA, which raises an enticing possibility that NS2-NS3pro cooperates with NS3hel to unwind double stranded RNA. To get insight into such possibility we took advantage of structural modeling proceeding through the following steps: a) building a model of RNA-NS2B-NS3pro complex; b) exploring relative positions and conformations of the NS2B-NS3pro and NS3hel domains; c) constructing a model of RNA bound to NS2B-NS3 complex; e) proposing a framework for double strand RNA unwinding including dynamic changes of NS2B-NS3pro conformations.

Close examination of three NS2B-NS3pro conformations revealed the presence of distinct 2 positively charged fork-like structures (hereafter referred to as forks) in the open conformation (**Fig 6**). Curiously, these structures were obstructed or misshaped in the closed and super-open conformations (**Fig 6**) making them prime candidates for RNA binding. Indeed, these forks were distinct, yet reminiscent of structures observed in single strand DNA binding protein from bacteriophage Enc34 (PDB ID: 5ODL) [36] and gp2.5 protein from bacteriophage T7 (PDB ID: 1JE5, 37]. We then modeled possible orientations of RNA bound by the NS3pro region encompassing these forks through the energy minimized using Amber18 [38] and ff14SB force field [39]. Such modeling provided a good volume fit and revealed a plausible position of single RNA strand readily accommodated by both forks (**Fig 6**). Note that such local structure (representing plausible RNA-binding cleft and positively charged forks) was

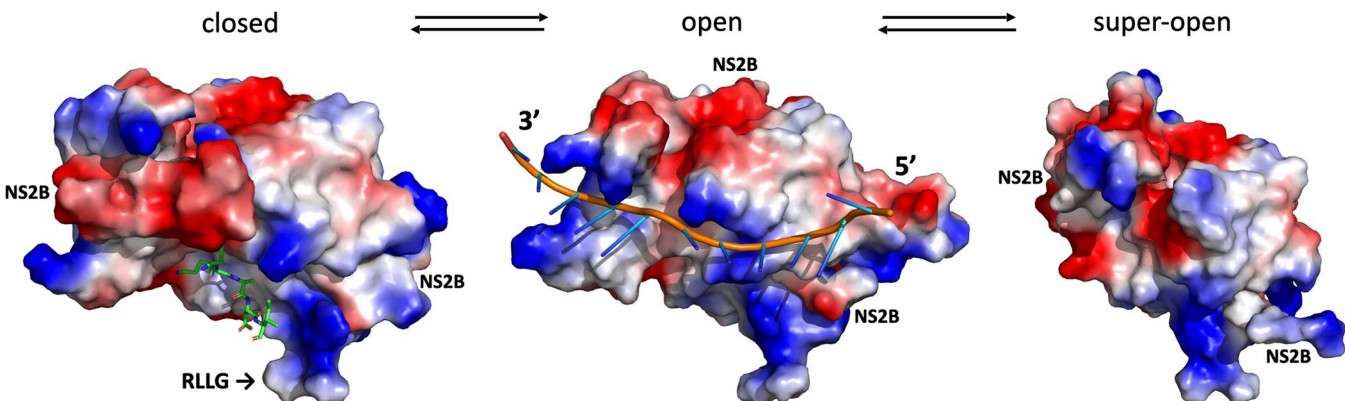

**Fig 6. The open conformation of ZIKV NS2B-NS3pro is uniquely suitable to bind RNA.** Closed (PDB ID: 5LC0), open (model based on WNV PDB ID: 2GGV) and super-open (PDB ID: 7M1V) conformation of ZIKV NS2B-NS3. The open conformation is shown with RNA inserted into the fork-like structures composed of positively charged amino acids (marked blue). One fork is close to 3'-end and another is located in the middle of the RNA strand. The structures are oriented so that RLLG loop of NS3pro is facing down. For the closed conformation a peptide-based substrate (RKADI, green ball and stick model) in the protease active center is modeled on the related structure of WNV NS2B-NS3pro + aprotinin (PDB ID: 2IJO).

either blocked by NS2B in the closed conformation or absent / disfigured in the super-open conformation. Our modeling demonstrates also that double stranded RNA is not able to bind open conformation of the NS3pro (S5 Fig). This is because the dsRNA is unable to provide continuous interactions between negatively RNA backbone and positively charged side chain amino acids in NS3pro.

## Modeling RNA-Helicase interactions

Next, we modeled RNA interactions with ZIKV NS3hel. We used the structure of ZIKV NS3hel bound to a short RNA fragment, AGAUC, (PDB ID: 5GJB [40]) to model a longer strand of the RNA by extending the short fragment along with structure optimization. We use bacterial helicase bound to single stranded DNA (PDB ID: 2P6R [41]) to build initial configuration of ZIKV helicase-RNA complex using 5GJB structure as a template (**Fig 7A**). This model contained a double helical part of RNA, before splitting and unfolding, oriented toward the 5'-end of the leading strand, and a single stranded RNA extending toward the 3'-end. This structure was then energy minimized using Amber18[38] and ff14SB force field [39].

## Modeling of NS3-pro-NS3-hel structure

To explore the flexibility of the linker between NS3pro and NS3hel domains we modeled possible mutual orientations of these domains focusing on NS2B-NS3pro in the open conformation and NS3hel built using three available crystal structure templates. Because the structure of ZIKV NS2B-NS3 (i.e. NS2B-NS3pro-NS3hel complex, protease and helicase together) is not available, we explored three crystallographic structures obtained from homologous MVE (PDB ID: 2WV9[42]) and DENV4 (PDB IDs: 2WZQ, 2WHX [21]) flaviviruses representing

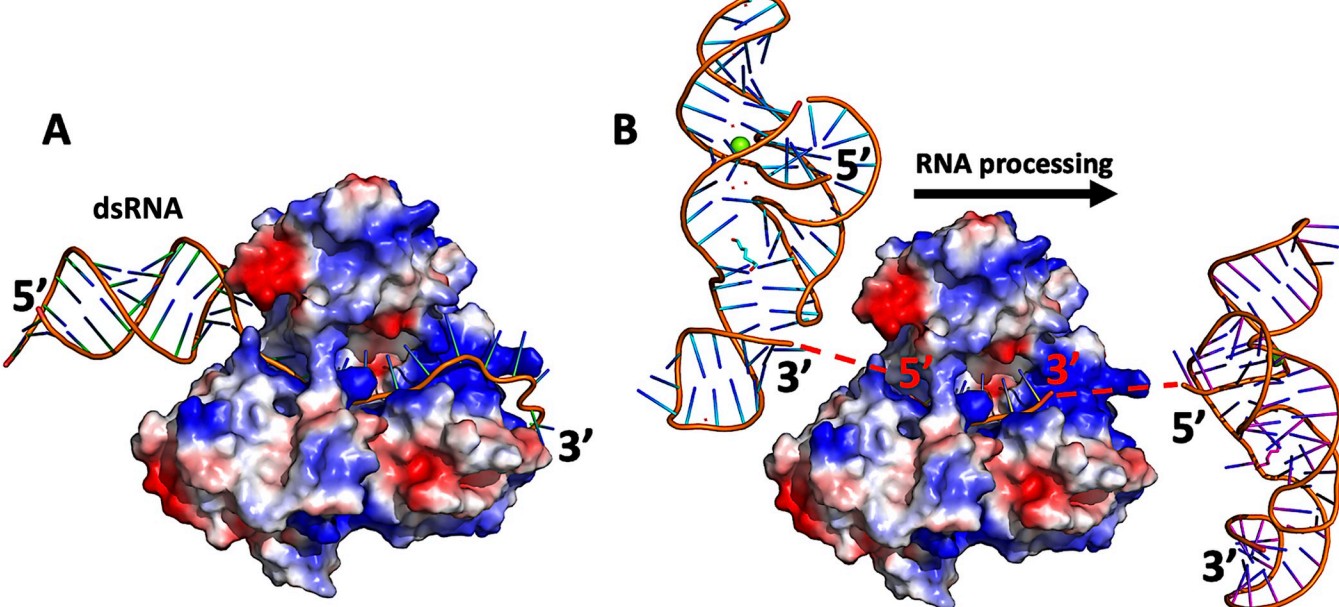

**Fig 7. Modeling directionality of RNA proceasing by ZIKV NS3hel.** (A) Panel represent a structure of ZIKV NS3hel with model build and AMBER force field minimized RNA fragment. On the left side of the molecule there is unprocessed dsRNA, while on the right side there is a leading strand of ssRNA after splitting and detaching the other strand. The ssRNA movement is facilitated by a patch of positively charged residues (right side of NS3hel–blue color). (B) Panel shows crystal structure (PDB id: 5GJB) of ZIKV NS3hel with short fragment of ssRNA (orange) and tentative positions of ZIKV RNA knots (PDB id: 5TPY). The modeling explains why RNA is processed from 3'-end toward 5'-end. Unwinding the knot is much more difficult if it is done via pulling its 5'-end (right side knot), due to stronger interaction with other nucleotides, then via pulling with its 3'-end (left side knot).

different mutual orientations of NS3pro and NS3hel domains. These available structures attest for an ample flexibility of the linker connecting the two domains. We built ZIKV NS3pro-NS3hel by structural alignment of ZIKV NS3hel (PDB ID: 5GJB) and the structures of NS2B-NS3pro in closed (PDB ID: 5lC0), open (our own model, based on 2GGV structure of NS2B-NS3-pro from WNV as a template) and super-open (PDB ID: 7M1V) conformations modeled onto MVE and DENV4 crystal structures. The results of modeling illustrate the wide range of motion available for NS3hel with respect to NS2B-NS3pro (**Fig 8A and 8C**). The mutual orientation of NS3pro and NS3hel that aligns the forks of NS3pro with the positively charged patches in NS3hel to facilitates RNA binding could be achieved by adjusting any of three models via rotation that involve only the movements of the flexible linker between NS3pro and NS3hel (**Fig 8D**). Therefore, the juxtaposition of the positively charged NS3pro forks and NS3hel patch in the open conformation of NS2B-NS3pro was selected as tentative RNA-bound configuration of NS2B-NS3 complex for further modeling.

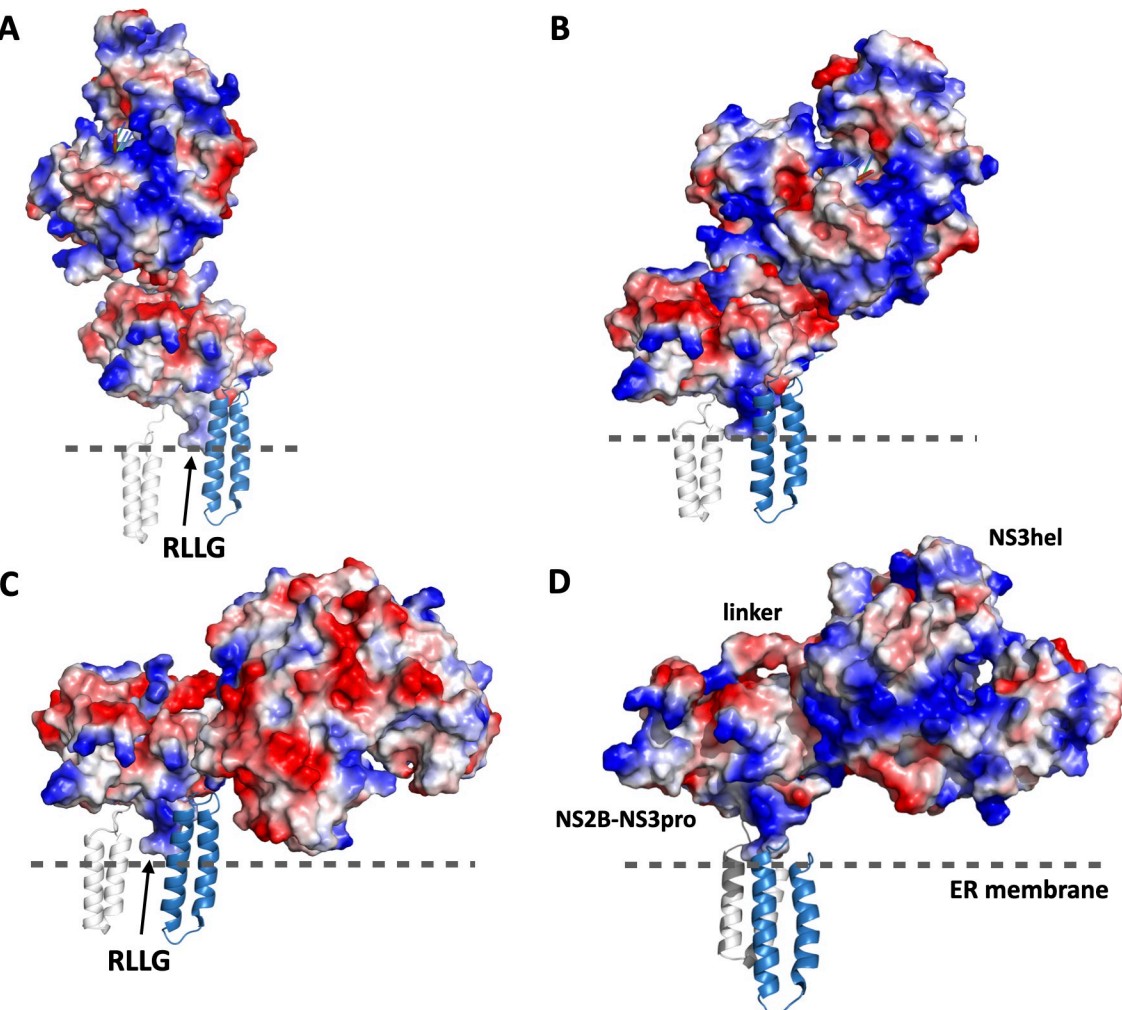

**Fig 8. Modeling ZIKV NS3pro-NS3hel mutual orientation.** Models of NS2B-NS3pro in the open conformation and NS3hel built using three different crystal structural templates demonstrate the flexibility of the linker between NS3pro and NS3hel. (A) 2WV9 structure template from MEV virus (PDB ID: 2WV9); (B) 2WHX structures structure template from DENV4 (PDB ID: 2WHX); (C) 2WZQ structure template from DENV4 (PDB ID: 2WZQ). (D) A homology-built model oriented to juxtapose the patch of positively charged residues in NS3hel domain and the positively charged forks in the open conformation of NS2B-NS3pro. The N- and C-terminal helices of NS2B (grey and sky-blue, respectively) are oriented to be inserted in the ER membrane denoted by the dashed line.

## Comparative analysis of RNA binding to NS2B-NS3 conformations

The juxtaposed model (**Fig 8D**) offers an appealing illustration of how RNA binds along the positively charged grove on NS3hel and the forks on NS3pro. We used this model to fit an extended strand of RNA by performing several steps of energy minimization with Amber18 [38] and ff14SB force field [39] resulting in a model of RNA-NS2B-NS3pro-NS3hel where the NS2B-NS3pro is in the open conformation (**Fig 9B**). To provide a more realistic view of the complex, we have oriented the N- and C- terminal helices of NS2B to be inserted in the ER membrane that was modeled to scale using a fragment of the membrane (PDB id: 2MLR [43]). The full model of NS2B has been built using the RoseTTAFold program. (**Fig 9**).

Similar arrangements were modeled for the closed (**Fig 9A**) and super-open (**Fig 9C**) conformations of ZIKV NS2B-NS3pro. Only in the open conformation the two fork patches of positively charged amino acids in NS3pro provided a plausible binding for the RNA strand. In the closed conformation, the RNA binding volume is occupied by negatively charged NS2B cofactor obstructing RNA binding (**Fig 9A**). In the super-open conformations, the RNA-binding area is distorted and the appropriate arrangement of positively charged forks could not be readily observed (**Fig 9C**).

## Conservation of amino acid sequences underlying RNA-binding forks across flaviviruses

We posit that if the fork-like structures present in ZIKV NS2B-NS3pro open conformation are functionally important for RNA-binding then their amino acid sequences will be conserved across flaviviruses. Indeed, the alignment of NS3pro from 11 flaviviruses revealed a near perfect amino acids identity / charge conservation for the forks 1–3 and to a lesser degree for the 4[th] fork (**Fig 10A**). Note that although not positively charged, the Asparagine (N) and Glutamine (Q) side chains are polar, known to readily form hydrogen bonds [44] potentially stabilizing the single RNA strand. The charge distribution that is clearly visible in the ZIKV RNA-NS2B-NS3pro model (**Fig 10B**) appears less pronounced for the WNV

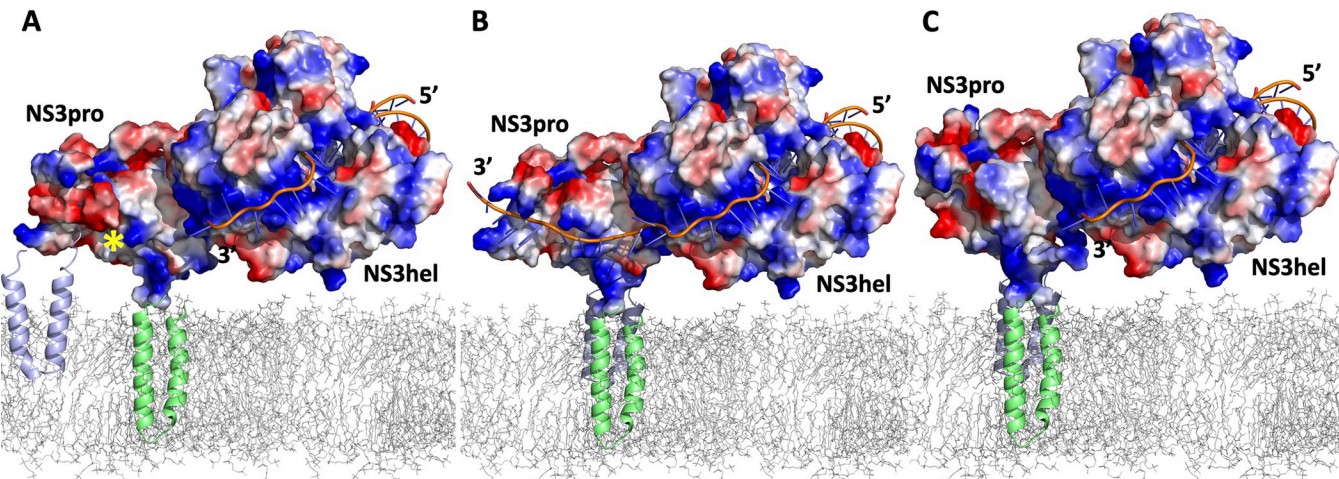

**Fig 9. Models of the RNA-NS2B-NS3pro-NS3hel complex.** The N- and C-terminal helices of NS2B (grey and green, respectively) are inserted in the ER membrane (grey mesh) modeled to scale using a fragment of the membrane (PDB ID: 2MLR). Similar orientations of NS2B-NS3pro with respect to NS3hel in (A) closed (PDB ID: 5LC0), (B) open (model based on WNV PDB ID: 2GGV), and (C) super-open (PDB ID: 7M1V) conformations. Position of the protease active site of NS2B-NS3pro in closed conformation is marked by yellow star. RNA strand in B is modeled to span a positively charged surface of NS3hel contiguous with positively charged forks in the open conformation of NS2B-NS3pro. Shorter RNA strands associated only with NS3hel domain is shown in A and C reflecting the lack of apparent RNA binding structures in the closed and super-open conformations of NS2B-NS3pro.

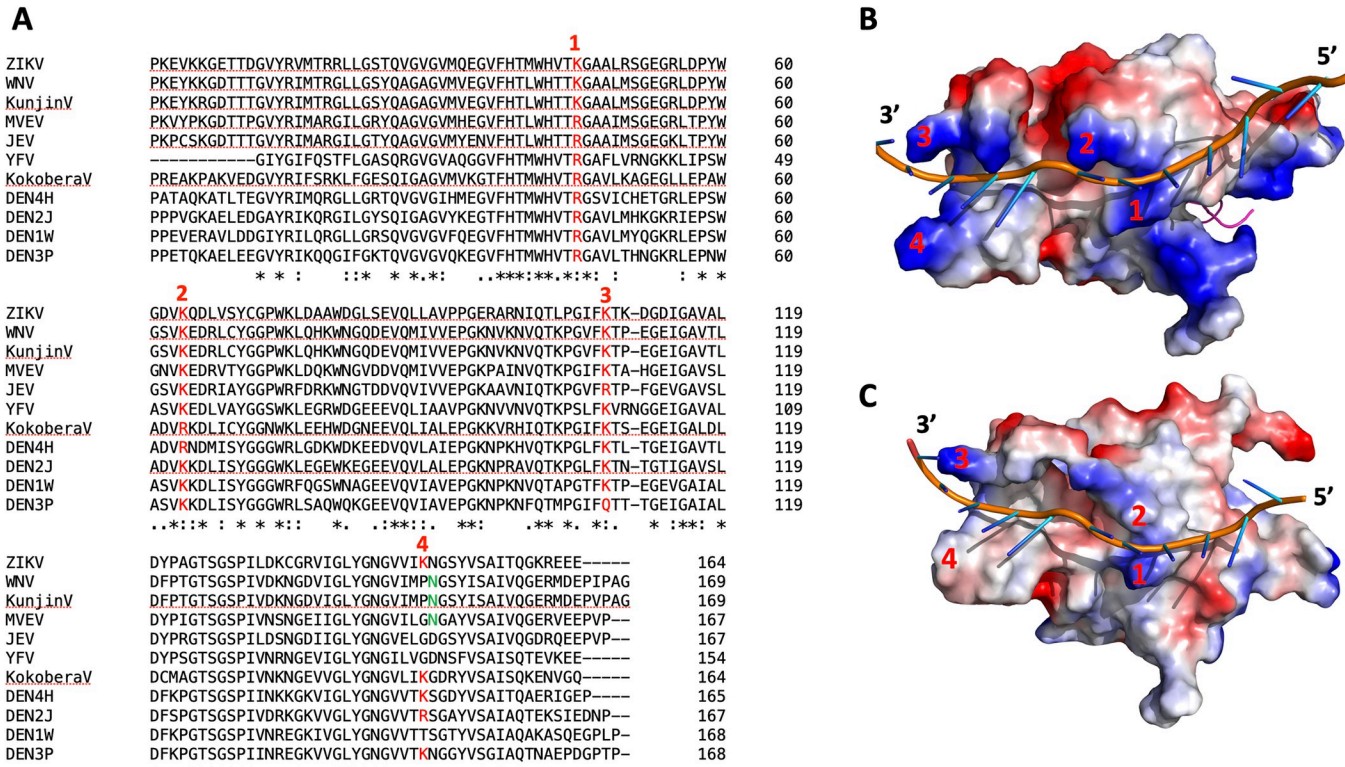

**Fig 10. Positively charged forks are conserved across flaviviruses.** (A) protein sequence alignment of NS3 from 11 flaviviruses. (B) A model of ZIKV NS2B-NS3pro in open conformation with the bound RNA. (C) A structure of WNV NS2B-NS3pro in open conformation (PDB id: 2GGV) with the bound RNA. Numbers in red (A through C) indicate corresponding conserved amino acids / structural positions of RNA-binding forks.

RNA-NS2B-NS3pro model (**Fig 10C**) in part due to the poorly resolved side chains in WNV NS2B-NS3pro structure (PDB ID: 2GGV).

## A "reverse inchworm" model

Based on our biochemical, structural, and modeling data we propose a novel model for a tightly intertwined NS2B-NS3 helicase-protease machinery of ZIKV (**Fig 11**). According to this model, a positively charged groove on the NS3hel surface accommodates the threaded single strand RNA and "hands it over" to NS2B-NS3pro. The open conformation of NS2B-NS3pro provides two positively charged/polar forks, contiguous with positively charged groove on NS3hel; these forks bind single strand RNA threaded by NS3hel (**Fig 11B**). Driven by ATP, NS3hel continues moving along and unwinding the double strand RNA which is a much larger molecule compared to NS3hel and is associated with several large viral proteins (e.g. RNA dependent RNA polymerase NS5). Because NS2B-NS3pro is permanently tethered to the ER membrane via NS2B's N-terminal and C-terminal membrane-associated domains and bound to single strand RNA, the NS3hel processivity results in the elongation of the linker between NS3pro and NS3hel domains (**Fig 11C**). At some point the linker is elongated to the maximum and NS3hel exerts a "yank" on NS3pro because NS3pro is tethered and NS3hel is moving. That force produces a change of NS2B-NS3pro conformation to the super-open and a release of the bound RNA strand (**Fig 11D**). The NS2B-NS3pro domain relaxes and then returns to the open conformation adjacent to NS3hel to enter the new cycle of RNA binding. This dynamic cycle of RNA binding and releasing enables the unlimited helicase processivity

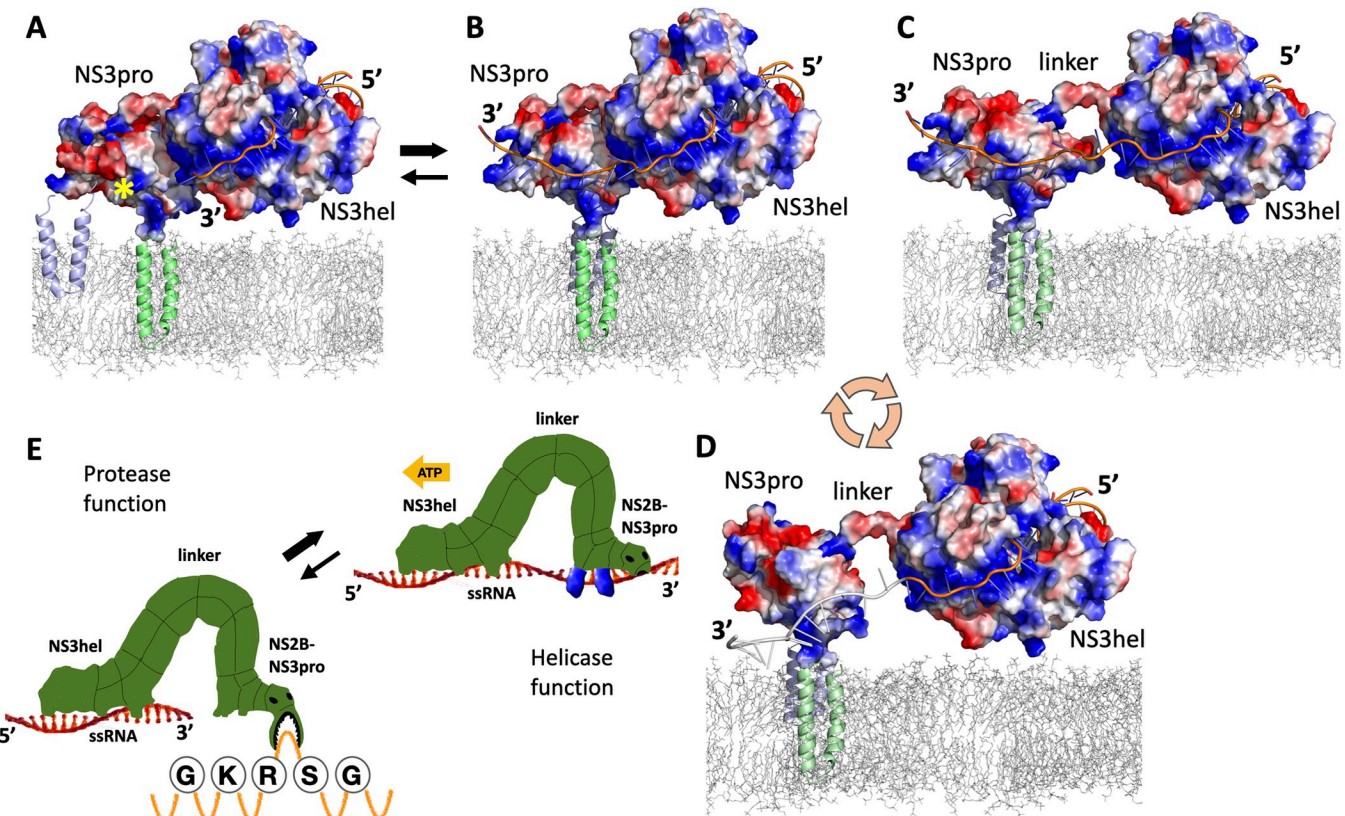

**Fig 11. A "reverse inchworm" model of the helicase-protease structure-activity cycle for ZIKV RNA-NS2B-NS3pro-NS3hel complex.** (A) NS2B-NS3pro is in the closed conformation (likely induced by a substrate), unable to bind RNA. Yellow star denotes the protease active center facing the ER membrane; (B) NS2B-NS3pro is in the open conformation (proteolytically inactive), binds RNA via 2 positively charged fork-like structures; (C) NS2B-NS3pro is switched to the open conformation with bound RNA, the linker between NS3pro and NS3hel domains is maximally elongated; (D) NS2B-NS3pro is in the super-open conformation unable to bind RNA; the dissociated RNA strand is cartooned in grey. The pink circular arrows indicate the cycle of NS2B-NS3pro conformational changes involved in dsRNA unwinding by NS3hel. (E) A cartoon of the reverse inchworm model showing the dissociated helicase and the protease activities of NS2B-NS3pro. The green "ATP" arrow indicates the direction of NS3hel movement unwinding dsRNA driven by ATP. GKRSS denotes a fragment of ZIKV polypeptide cleaved by ZIKV Ns2B-NS3pro between Arg (R) and Ser (S). All models built to scale based on the crystal structures and energy minimization (see text for details).

along the ~11 kb of dsRNA flavivirus genome. The protease activity requires a distinct closed conformation of NS2B-NS3, which is most likely induced by the presence of a substrate such as peptide loops extended from the ER membrane (**Fig 11A**). Note that the reverse inchworm model could be readily extended to all flaviviruses given a very high degree of sequence and structure conservation of NS2B-NS3 complex within the family.

## Novel approach to targeting ZIKV NS2B-NS3 functions

We speculate that a particular spatial arrangement of NS2B-NS3pro and NS3hel enabling RNA binding to both domains simultaneously via positively charged elements in both NS3pro and NS3hel could be an important element in the viral replication cycle. Our model suggests that targeting such specific alignment with small molecules (e.g. dual binding molecules that bridge both domains) could be a promising therapeutic strategy. Such small molecules could fit into the cavities located at the interface between NS3pro and NS3hel. We thus evaluated the sizes and locations of these cavities on the interface of ZIKV NS2B-NS3pro and NS3hel with the POVME3.0 software [45]. The volume measurements are in the range of 150–680 Å for

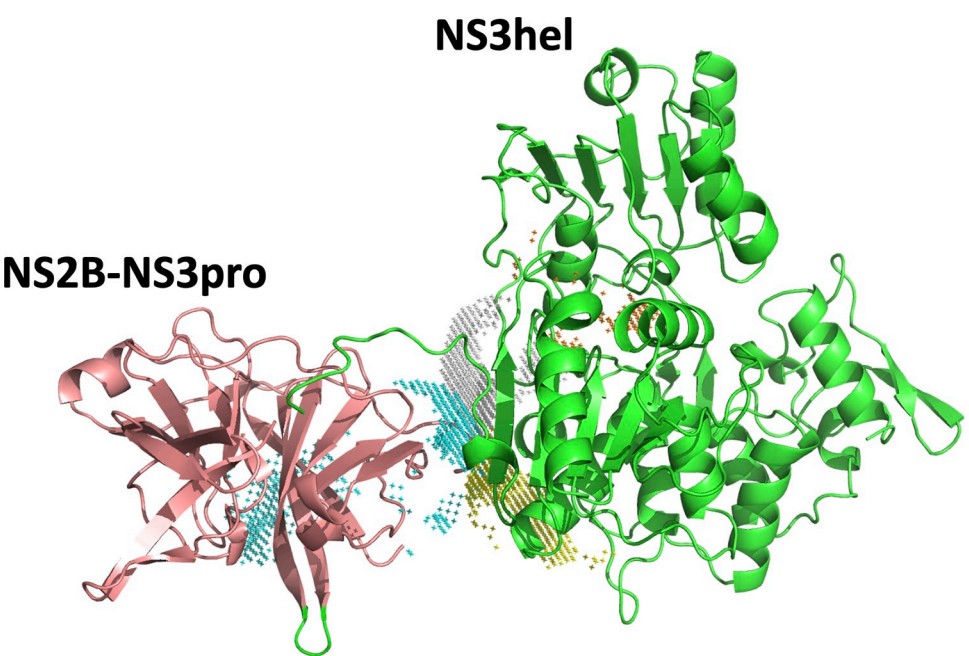

**Fig 12. Volumes and position of the cavities at the NS3pro and NS3hel interface.** Positions of the cavities at the interface of the best aligned NS3pro and NS3hel to accommodate single stranded RNA (same as Fig 8D). Colored meshed areas mark the positions of cavities.

NS3hel and 300–650 Å for NS3pro, thus a combined volume of 500–1300 Å is available to accommodate a dual binding molecule (**Fig 12**). For comparison, the volume of ATP molecule is ~650 Å. Therefore, both individual and combined volumes of such cavities could accommodate small molecules that could be designed to interfere with this specific alignment of NS2B-NS3pro and NS3hel.

### Potential caveats and Interpretations

The RNA-binding ZIKV NS2B-NS3pro construct was purified *via* the His tag prior to the thrombin-mediated removal of the tag. In principle, other *E. coli* proteins could have been co-purified during affinity chromatography affecting RNA binding. For instance, the heat-stable *E. coli* Hfq protein is known to harbor stretches of His residues which could be co-purified during Ni affinity chromatography resulting in false-positive RNA-binding activities [46]. However, our purification scheme involved thrombin cleavage of the His tag to release ZIKV NS2B-NS3pro, and the thrombin site is absent in *E. coli* Hfq. Further, Hfq is an 11 kDa protein and has $IC_{50}$ values for RNA binding in the low nanomolar range, which is inconsistent with our results for NS2B-NS3pro binding to RNA with $IC_{50}$ value in the sub-micromolar range. Critically, RNA binding to NS2B-NS3pro was abrogated by aprotinin, WRPK3, and several allosteric inhibitors of NS2B-NS3 ruling out the involvement of Hfq. Additional experiments will be required to formally prove that the open conformation of ZIKV NS2B-NS3pro binds to RNA. One definitive approach will be to crystallize the RNA-NS2B-NS3pro complex; such efforts are underway.

### Discussion

In all flaviviruses, the proteolytic activity of NS2B-NS3pro and the helicase activity of NS3hel are constrained within a single NS3 polypeptide. Extensive structural and functional studies

have conclusively demonstrated a strong interdependence in the enzymatic activities of the protease and helicase domains of flaviviral NS3 polypeptide [18,19,47,48]. Multiple studies have pointed to the critical role of the flaviviral NS3pro domain for the proper functioning of NS3 helicase [18–20]. The protease domain of HCV NS3 protein plays a significant role in the stepping efficiency of NS3hel by increasing the efficiency of the coupling construct NS3-NS4A up to 93% per each ATP hydrolysis event relative to 20% for a single NS3 helicase construct [18]. The presence of the NS3pro domain within full-length HCV NS3 protease-helicase constructs was shown to significantly enhance RNA binding to NS3 helicase [19]. The absence of the DENV2 NS3pro domain results in the malfunctioning of DENV2 NS3 helicase directional movement and the appearance of two opposing activities characterized by an ATP-dependent steady state between RNA unwinding and RNA annealing processes [20]. The DENV4 virus full-length NS3 protein displayed a 10-fold higher ATP affinity than the isolated helicase domain [49]. Moreover, for the DENV2 virus NS3 helicase-protease construct, the RNA unwinding activity was about 30-fold higher than that of the NS3hel construct representing only helicase domain [50]. Also, a higher rate of RNA unwinding was reported for Kunjin virus (KUNV) full-length NS3 protein when compared to a construct containing only NS3hel domain [51]. Despite the robust evidence that NS3pro is important for the helicase function, the mechanism of the NS2B-NS3pro involvement in NS3 helicase activity remains obscure.

Our discovery that RNA binds only the open conformation of ZIKV NS2B-NS3pro enabled us to formulate a novel "reverse inchworm" model for a tightly intertwined NS2B-NS3 helicase-protease machinery of ZIKV and flaviviruses in general. Critically, our model provides a previously unappreciated mechanistic connection between closed, open, and super-open conformations of NS2B-NS3pro reported for many flaviviruses. The key element of this model is a catch-and-release cycle where NS2B-NS3pro binds RNA in the open conformation and releases RNA in the super-open conformation. Although reminiscent of the "inchworm" mechanism proposed for PcrA DNA helicase [52], the "reverse inchworm" model has several principal distinctions. First, in the case of PcrA DNA helicase, the ATP hydrolysis drives an alternation in affinity for ssDNA of domains 1A and 2A, thus directly promoting DNA strand translocation through the catch-and-release cycles [52]. In the case of ZIKV NS2B-NS3 RNA helicase, the catch-and-release cycles are driven by a conformational switch uncoupled from the immediate result of ATP hydrolysis. Second, the NS3pro is anchored into the ER membrane through NS2B while being connected by a flexible linker to NS3hel, effectively making these 2 RNA binding sites much more independent compared to the closely opposed A1 and A2 ssDNA binding site of PcrA. Third, the major distinction of the ZIKV protease-helicase is that NS2B-NS3pro functions as a protease only when it adapts a closed conformation. The entire spectrum of NS2B-NS3pro-NS3hel activities naturally evokes the image of a reversing inchworm (**Fig 11E**).

The binding of single-stranded RNA in our model is mediated by two positively charged fork-like structures present only in the open conformation of NS2B-NS3. The amino acid sequences and 3D arrangements of such forks are highly conserved across flaviviruses suggesting a universal nature of the proposed mechanism. The lack of NS3pro results in a major reduction of the flaviviral helicase activity [21,22,53] possibly because the NS3hel lacks the directionality of RNA threading upon ATP hydrolysis akin to that of bi-depictional RNA helicases [54–57]. In this scenario, the backward movement of NS3hel is blocked by NS3Pro bound to single-stranded RNA thus enforcing the directionality of NS3hel towards double-stranded RNA. Further structural work (e.g. crystallization of the RNA-NS2B-NS3pro-NS3hel complex) will be required to elucidate how exactly single-strand RNA binds to such structures within NS3pro in different flaviviruses.

The crystal structure of Japanese encephalitis virus (JEV) NS2B-NS3pro (PDB ID 4R8T) was uncovered in 2015 [24]. However, no structural or functional analysis was reported for this crystal structure, likely explaining the lack of attention paid by the research community. Note that polypeptides with over 35% identity are very likely to have a similar fold [58]. Given over 50% identity between flaviviral proteases across the family [23,59], we posit that the super-open conformation demonstrated for JEV and ZIKV NS2B-NS3pro is a common feature of the Flaviviridae family.

Previously, two ZIKV NS2B-NS3pro constructs, eZiPro [60] and bZiPro [61], were crystallized in the closed conformation. Indeed, in both cases, the active center of ZIKV NS2B-NS3pro is occupied by a short peptide fragment, likely inducing such closed conformation. We superimposed eZiPro (PDB ID 5GJ4) with bZiPro (PDB ID 5GPI) to better demonstrate that the active center in both structures is occupied either by tetrapeptide TGKR (T127-G128-K129-R130) originating from the NS2B C-terminus (eZiPro) or by a tetrapeptide KKGE (K14-K15-G16-E17) originating from a neighboring NS3 molecule (bZiPro) (**S5 Fig**). To the best of our knowledge, there are no crystal structures of flaviviral NS2B-NS3 proteases in the closed conformation without peptide/inhibitor in the active center. We take it as an indication that the closed conformation is always induced when a substrate is present in the active center. Curiously, the $^{15}$N $R_2$ NMR signal from NS2B residues 65–85 is missing in bZiPro alone but re-appears when AcKR is added [61]. This is consistent with the idea that without AcKR, bZiPro exists in the open conformation where much of the C-terminal part of NS2B is dissociated from NS3Pro and remains unstructured, thus resulting in the lack of NMR signal.

Organelle-like alterations of ER membranes called replication factories (RF) are associated with ZIKV / flaviviral infections. The detailed architecture of RFs has been reconstructed using electron tomography [62–67]. Replication Factories consist of several sub-structures, including vesicle packets (VP) and virus bags (newly assembled virions) which are morphologically distinct [62–67]. According to this model, NS2B-NS3 complex will function mostly as a helicase within VPs and mostly as a protease outside VPs since ribosome clusters indicative of translation areas were visualized adjacent to but outside of VP structures [68]. The RF dynamics of ZIKV is compatible with the reverse inchworm model of NS2B-NS3 functional cycles. However, how the intact ~11kB viral RNA is transported out of VPs and loaded on the ribosomes remains enigmatic.

In a recent study, Xu et al., 2019 [53] suggested that ZIKV NS3 is a canonical RNA helicase but observed a very limited processivity of recombinant NS3 polypeptide that was capable of unwinding no longer than 18 bp RNA duplex. Curiously, the addition of NS5 polymerase increases the speed but doesn't change the processivity (maximum 18 bp RNA duplex) of NS3 [53], suggesting that limitation is intrinsic to NS3. The recombinant NS3 polypeptide used in Xu et al., 2019 study comprised both NS3pro and NS3hel; in fact, recombinant NS3hel alone showed no helicase activity whatsoever. However, that single NS3 polypeptide lacks NS2B. Note that NS2B is always tightly associated with NS3pro via a three-strand beta-barrel (aa 49–58 of NS2B), which remains intact in all NS3pro conformations. Both the N-terminal and C-terminal ends of NS2B are anchored into the ER membrane, therefore, tethering the entire NS2B-NS3pro-NS3hel complex to the ER membrane. In the absence of an NS2B cofactor, NS3pro could still adopt an open-like conformation capable of RNA binding. However, without the NS2B cofactor firmly anchoring NS3pro into the ER membrane, upon maximum elongation of the 12 aa linker, NS3hel wont be ableto "yank" on NS3pro. Therefore, the switch to the super-open conformation, which forces RNA dissociation, was missing. The results of Xu et al., 2019 are thus consistent with our reverse inchworm model, which also predicts that the addition of the membrane-anchored form of NS2B cofactor, NS2B-NS3 complex, will exhibit

unlimited processivity needed to dissociate ~11kb of the entire ZIKV dsRNA. Finally, it is likely that components of NS2, NS3, NS4, and NS5 polypeptides work in concert as one coordinated complex where various subunits of NS2, NS4A and NS4B proteins may provide sophisticated anchoring of the entire complex to the ER membrane. Indeed, such a complex has recently been proposed [69].

The 12 aa linker between the NS3pro and NS3hel domains was shown to be critical for the protease and helicase activities in the structurally and functionally similar DENV2 NS2B-NS3pro[21]. Mutations in the linker affecting its flexibility and length were crucial for the ATPase and helicase activities and led to significant reductions in viral genomic RNA synthesis [21]. The linker is negatively charged due to 5 glutamic acid residues spread along the sequence ([171]EEETPVECFEPS[182]). In crystal structures, the ZIKV protease-helicase linker is disordered (PDB IDs 5TFN, 5TFO, 5T1V and 6UM3). However, in the closed and open structures of WNV and DENV2 proteases, the linkers are associated with a conserved region of the protease domain [23]. We propose that this flexible negatively charged peptide competes with RNA for binding to NS2B-NS3pro thus providing an explanation for the observed lack of RNA binding by the NS2B-NS3pro-long construct that includes the 12 aa linker.

Finally, the reverse inchworm model posits that NS3hel "hands over" the threaded RNA strand to the NS2B-NS3pro. This process is the most efficient when the NS3hel positively charged groove is contiguous with the positively charged fork-like structures in the open conformation of NS2B-NS3pro. The volumes of the cavities formed by closely opposed ZIKV NS3pro and NS3hel domains are consistent with the possibility of designing small molecules that would bind on one or both sides, thus interfering with this arrangement. We propose that such precise orientation of NS3pro and NS3hel domains is critical for the most efficient RNA processing by NS3 helicase. We put forward the idea that targeting the interface of the specific alignment of NS3pro and NS3hel domains proposed here with bivalent small molecules that can bridge NS3pro and NS3hel effectively freezing this transient conformation could be a universal and highly specific approach targeting the replication of ZIKV and flaviviruses in general.

## Materials and methods

### Reagents

Routine laboratory reagents were purchased from Millipore Sigma (St. Louis, MO) unless indicated otherwise. Horseradish peroxidase-conjugated donkey anti-mouse IgGs and TMB/M substrate were from Jackson ImmunoResearch Laboratories (West Grove, PA), SuperSignal West Dura Extended Duration Substrate for ECL was from ThermoFisher (Carlsbad, CA) and SurModics IVD (Eden Prairie, MN), and oligonucleotides were synthesized by Integrated DNA Technologies (San Diego, CA).

### Cloning, expression, and purification of ZIKV NS2B-NS3pro constructs

DNA sequences for the ZIKV constructs were synthesized and codon-optimized for efficient transcription in *E. coli*. The constructs were designed as single-chain two-component products lacking the hydrophobic transmembrane domain of the NS2B cofactor. To achieve this, the cytoplasmic portion of NS2B (residues 48–94) was linked to the NS3pro domain *via* a 9-residue linker (GGGGSGGGG). To block the proteolytic activity of the construct, where indicated, the catalytic Ser[135] was substituted with Ala to give the inactive Ser135Ala construct.

ZIKV NS2B-NS3pro recombinant constructs with N-terminal His tag were used to transform competent *E.coli* BL21 (DE3) Codon Plus cells (Stratagene). Transformed cells were grown at 30˚C in LB broth containing carbenicillin (0.1 mg/ml). Protein production was

induced with 0.6 mM IPTG for 16 h at 18˚C. Cells were collected by centrifugation at 5000 g at 4˚C, and the cell pellet was resuspended in 20 mM Tris-HCl buffer, pH 8.0, containing 150 mM NaCl (TBS), and sonicated (eight 30 s pulses at 30 s intervals) on ice. The sample was then centrifuged at 40,000 g for 30 min at 4˚C and the constructs were purified from the supernatant fraction using Ni-NTA Sepharose equilibrated with TBS containing 1 M NaCl. Impurities were removed by washing with the same buffer supplemented with 35 mM imidazole, and the column was equilibrated with standard TBS. The beads were co-incubated with thrombin (Sigma Aldrich) to cleave the His tag and release NS2B-NS3pro. Fractions containing recombinant protein were combined and purified by gel filtration on a S200 26/60 column (GE Healthcare) equilibrated with standard TBS. Fractions containing ZIKV NS2B-NS3pro were concentrated to approximately 10 mg/ml using 10 kDa-cutoff concentrators (Millipore, Billerica, MA), and then flash frozen in small aliquots and stored at –80˚C. Purity of the material was checked by SDS-PAGE (12% NuPAGE-MOPS, Invitrogen) followed by Coomassie staining.

We generated a series of constructs (Mut3, Mut5, and Mut6) that forced ZIKV NS2B-NS3pro to adopt the "super-open" conformation using the wild-type construct as a template. In the Mut3 construct, all three Cys residues in the original sequence were substituted with Ser residues (Cys80Ser, Cys143Ser, and Cys178Ser). In the construct with the "super-open" conformation only (Mut5), two additional Cys residues were inserted into the Mut3 construct (Ala88Cys and Lys157Cys). To obtain a catalytically inactive mutant with the "super-open" conformation (Mut6), an additional Ser135Ala was introduced into the Mut5 construct. For crystallization purposes, we also created a Mut7 construct containing Leu30Thr and Leu31Ser mutations in the Mut5 sequence. Mut7 was designed to minimize ZIKV protease dimerization during crystallization.

The DNA constructs were cloned into pGEX6P1 plasmid using BamH1 and EcoR1 cleavage sites, resulting in fusion of a GST tag at the N-terminus of the NS2B cofactor. *E. coli* were transformed with the individual recombinant constructs, and protein production was induced, and the cells were disrupted as described above. The supernatant fraction containing the GST-tagged constructs were loaded onto Protino Glutathione Agarose 4B (Fisher Scientific) beads and impurities were removed by washing with TBS. To cleave the GST tag from the viral protease, the beads were co-incubated with 3C protease (Genscript). The NS2B-NS3pro constructs were then additionally purified by gel filtration on an S200 26/60 column (GE Healthcare) equilibrated with TBS. Purity of the purified construct was analyzed using SDS-PAGE followed by Coomassie staining. Purified protease was concentrated to approximately 10 mg/ml using 10 kDa-cutoff concentrators (Millipore, Billerica, MA), flash frozen in small aliquots, and stored at –80˚C. To isolate NS2B-NS3pro complexed with aprotinin or WRPK3, the NS2B-NS3pro constructs were incubated with each inhibitor at an equimolar ratio and the protein-inhibitor complexes were purified by gel filtration on a S200 Superdex column.

### Fluorescent proteinase activity assay and $IC_{50}$ determination

The peptide cleavage activity assays with purified ZIKV NS2B-NS3pro polypeptides were performed in 0.2 ml TBS containing 20% glycerol and 0.005% Brij 35 containing 20 μM of the cleavage peptide pyroglutamic acid Pyr-Arg-Thr-Lys-Arg-7-amino-4-methylcoumarin (Pyr-RTKR-AMC) and 10 nM enzyme. The reaction velocity was monitored continuously at $\lambda_{ex}$ = 360 nm and $\lambda_{em}$ = 465 nm on a Tecan fluorescence spectrophotometer (Männedorf, Switzerland). To determine the $IC_{50}$ values of the inhibitory compounds, ZIKV NS2B-NS3pro constructs (20 nM) were preincubated for 30 min at 20˚C with various concentrations of compounds in 0.1 ml TBS containing 20% glycerol and 0.005% Brij 35. Pyr-RTKR-AMC

substrate (20 μM) was then added in 0.1 ml of the same buffer. $IC_{50}$ values were calculated by determining the compound concentration required to obtain 50% of the maximal inhibition of NS2B-NS3pro activity against Pyr-RTKR-AMC. GraphPad Prism was used as fitting software. All assays were performed in triplicate in 96-well plates. The same catalytic assays were used for WNV, and DENV proteases as previously described [13,14,22,23].

## Fluorescence polarization assay of RNA/ssDNA binding to ZIKV NS2B-NS3 protease

Binding between purified recombinant ZIKV NS2B-NS3pro constructs and either RNA or ssDNA was assessed using a fluorescence polarization (FP) assay conducted in 0.1 ml of TBS containing 1 mM $MgCl_2$ at 25˚C for 1 h. Samples of 10 nM 6-carboxyfluorescein-labeled 20-base RNA (poly-rA or -rU) or ssDNA (poly-dA or -dT) oligonucleotides were incubated with 50 nM to 50 μM of purified NS2B-NS3pro constructs. Polarization was monitored on a **Bruker Daltonics** fluorescence spectrophotometer (Fremont**, CA**). $K_d$ values for RNA and ssDNA binding were calculated by determining the construct concentration needed to reach 50% polarization for 3′-fluorescein amidite-labeled RNA/DNA. All assays were performed in triplicate in 96-well plates. GraphPad Prism was used as fitting software.

## Molecular modeling

All structural modeling presented here used the FFAS sequence alignment server for finding structural templates [25] in Protein Data Bank and the MODELLER program for creating homology models [26]. For building a structural model for the full protein sequence, e.g., ZIKV NS2B co-factor protein, we employed RoseTTAFold program that uses deep learning approach for quick structure prediction [27]. Volumes and position of the cavities at the NS3pro and NS3hel interface calculated using POVME3.0 program [45].

## Supporting information

**S1 Fig. ZIKV polyprotein composition and processing by viral and host cell proteases.** (A) Positions of cleavage sites for host and viral proteases at the junctions between individual viral proteins are indicated by arrows. (B) NS2B-NS3pro cleavage sequences in flaviviral polyproteins. Cleavage sites in the capsid protein C and at the NS2A/NS2B, NS2B/NS3, NS3/NS4A, NS4A/NS4B, and NS4B/NS5 boundaries are shown. ZIKV, Zika (GenBank AMB37295); WNV, West Nile virus (GenBank P06935); JEV, Japanese encephalitis (GenBank P19110); YFV, yellow fever (GenBank P19901); DENV1–4, dengue serotypes 1–4 (GenBank P33478, P29990, P27915, and P09866, respectively).
(PDF)

**S2 Fig. An overlap of ZIKV NS2B-NS3pro (PDB ID 7M1V) and JEV NS2B-NS3pro (PDB ID 4R8T).** Color scheme: pink and blue—NS3pro and NS2B, respectively, from ZIKV. Light grey and dark grey—NS3pro and NS2B from JEV. Red color marks catalytic residues in ZIKV structure. Note a perfect overlap of all structurally resolved elements. The divergent tails of NS2B (blue and dark grey) were not resolved in either structure.
(PDF)

**S3 Fig.** (A) NS2B-NS3pro constructs used in this study. All constructs were N-terminally fused with GST protein or HisTag for purification. NS2B central hydrophilic portion is shown in blue and the NS3 protease in green. L = GGGGSGGGG linker between NS2B and NS3pro. (B) Western blot analysis of purified wild-type and Mut7 NS2B-NS3pro proteins.
(PDF)

**S4 Fig. Superimposition of eZiPro5 and bZiPro6 crystal structures.** A. eZiPro (PDB ID GJ4), NS3pro and NS2B are marked with light and dark blue, respectively. Peptide fragment TGKR (bound to eZiPro) shown in green. B. bZiPro (PDB ID 5GPI), NS3pro and NS2B are marked with magenta and grey, respectively. Peptide fragment KKGE (bound to bZiPro) shown in orange. C. Superposition of A and B. Catalytic residues marked in red. Note that protease catalytic center is occupied in both crystal structures.
(PDF)

**S5 Fig. Superimposition of eZiPro5 and bZiPro6 crystal structures.** A. eZiPro (PDB ID GJ4), NS3pro and NS2B are marked with light and dark blue, respectively. Peptide fragment TGKR (bound to eZiPro) shown in green. B. bZiPro (PDB ID 5GPI), NS3pro and NS2B are marked with magenta and grey, respectively. Peptide fragment KKGE (bound to bZiPro) shown in orange. C. Superposition of A and B. Catalytic residues marked in red. Note that protease catalytic center is occupied in both crystal structures.
(PDF)

## Acknowledgments

We thank Dr. Alexander Aleshin for critical discussions and suggestions and for providing early access to PDB ID 7M1V. We thank Dr. Alex Strongin for valuable advice and discussions, and Dr. Sumit Chanda for critical support to S.A.S. and valuable discussions.

## Author Contributions

**Conceptualization:** Sergey A. Shiryaev, Robert C. Liddington, Alexey V. Terskikh.

**Formal analysis:** Sergey A. Shiryaev, Piotr Cieplak, Anton Cheltsov.

**Funding acquisition:** Robert C. Liddington, Alexey V. Terskikh.

**Investigation:** Sergey A. Shiryaev, Piotr Cieplak, Alexey V. Terskikh.

**Methodology:** Sergey A. Shiryaev, Piotr Cieplak, Alexey V. Terskikh.

**Project administration:** Alexey V. Terskikh.

**Software:** Piotr Cieplak, Anton Cheltsov.

**Supervision:** Alexey V. Terskikh.

**Writing – original draft:** Sergey A. Shiryaev, Alexey V. Terskikh.

**Writing – review & editing:** Sergey A. Shiryaev, Alexey V. Terskikh.

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
