## [Decision Letter · Decision Letter 0]

27 Sep 2023

Dear Dr Terskikh,

Thank you very much for submitting your manuscript "DUAL FUNCTION OF ZIKA VIRUS NS2B-NS3 PROTEASE" for consideration at PLOS Pathogens. As with all papers reviewed by the journal, your manuscript was reviewed by members of the editorial board and by several independent reviewers. The reviewers appreciated the attention to an important topic. Based on the reviews, we are likely to accept this manuscript for publication, providing that you modify the manuscript according to the review recommendations.

Please pay particular attention to the requests of Reviewer #2. The critique here includes that the revised manuscript needs re-organization in order for a reader to follow the work, and considerable effort was made by the Reviewer in aiding this process.

Sincerely,

Sonja M. Best, Ph.D.

Section Editor

PLOS Pathogens

Sonja Best

Section Editor

PLOS Pathogens

Kasturi Haldar

Editor-in-Chief

PLOS Pathogens

orcid.org/0000-0001-5065-158X

Michael Malim

Editor-in-Chief

PLOS Pathogens

orcid.org/0000-0002-7699-2064

Reviewer Comments (if any, and for reference):

Reviewer's Responses to Questions

**Part I - Summary**

Reviewer #1: This study provides evidence to exhibit the novel/dual function of NS2B-NS3, giving new understanding to this complex, which might provide a strategy in antiviral development. Quite a few methods have been applied to evaluate their observation. Their reply to the reviewers' questions seems to be reasonable.

Reviewer #2: Most of the comments from the previous review were addressed adequately where possible. By and large, the authors did a good job at incorporating the requested questions and ideas into their discussion of their work. However, there were a couple comments that should be further addressed:

- A negative control for Figure 4A: While the in silico modelling of dsRNA is compelling, this should either be included in the manuscript (in Figure 4 or supplemental) or the authors should perform the curve for dsRNA. This point needs to be directly addressed in the manuscript to strengthen the authors discovery of a ssRNA binding channel.

- The authors should further consider/discuss the path of the RNA from dsRNA unwinding to (+) strand synthesis and the role of NS5 to refine their model.

In addition, a number of the new additions to address the reviewer’s previous comments should be reorganized into the existing figures or moved to the supplemental figures, with the text consolidated and condensed accordingly (see more detailed thought in the minor comments section below).

**Part II – Major Issues: Key Experiments Required for Acceptance**

Reviewer #1: The binding between RNA and NS2B-NS3 observed in this study is novel, but confirming the interactions inside cells is challenging. The application of artificial constructs in the binding in the is not perfect, but the conclusion is supported by the data. The observation in this study is useful for further studies.

Reviewer #2: - A negative control for Figure 4A: While the in silico modelling of dsRNA is compelling, this should either be included in the manuscript (in Figure 4 or supplemental) or the authors should perform the curve for dsRNA. This point needs to be directly addressed in the manuscript to strengthen the authors discovery of a ssRNA binding channel.

**Part III – Minor Issues: Editorial and Data Presentation Modifications**

Reviewer #1: Authors tried to address my questions. Some minor parts:

>please indicate the RLLG sequence in Figure 8 to have a better view of the model.

>The authors believe that the substrate will dissociate with protease cleavage site under the native conditions. The binding affinity between the TGKR sequence and protease is in mM range (PMID: 29908184), which is weaker than that of RNA sequence. Authors should consider adding some discussion on this, e.g. explain the possible binding inside cells.

Reviewer #2: Comments

- A number of the new sections seemed slightly out of place with respect to the organization of the manuscript. For example:

o i) The “Super-open conformation is conserved between ZIKV and JEV” doesn’t feel sufficient to be its own section, and might be better served by being integrated into the following section to provide support for it.

o ii) Figure 7 seems like it should be supporting one of the other figures (as a panel) at the point where the authors directly tie in helicase unwinding of dsRNA (perhaps Figure 9 or 10?).

o iii) Figure 8 could be a supplemental figure to Figure 9 (couldn’t see in the file, but by assumption) to support the conformations of the 2B-3prot in the context of the full NS3 protein and the membrane. The main output orientation from this figure is already included in Figure 11.

o iv) Figure 10 could be an effective panel in Figure 9 to directly support the forks.

o v) Figure 12 should be in the discussion, rather than the results. Figure 11 may also be well-served by being moved to the discussion and reorienting the discussion more so around the model.

- Introduction: the following statement: “Recently, we determined the crystal structure of ZIKV NS3pro with the covalently linked NS2B cofactor with 2.5 Å and 3 Å resolution” - is missing a reference, unless the authors mean herein? (unclear). If this is referring to herein, this belongs in the results. The paper referred to in this paragraph is not from the authors.

- Figure 1: for the non-structural biologist, indicating that blue is positive-charge and red is negative-charge would be helpful in the figure legend.

- Results: section entitled “Super-open conformation is conserved between ZIKV and JEV” – should be incorporated into the following section as it doesn’t really bring a new result on its own. Rather belongs with (in support) of the subsequent modeling.

- Results: section entitled: “RNA binding inhibits the proteolytic activity of ZIKV NS2B-3pro” – second sentence should read “…adopts an open proteolytically inactive conformation.”

- In the last sentence of the “RNA binding inhibits the proteolytic activity of ZIKV NS2B-NS3pro” section, the authors state that “The observed inhibition of proteolytic activity is likely due to RNA-induced stabilization of NS2B-NS3pro in the open conformation, which is proteolytically inactive.” A reference (or reiterating of a reference) should be used to support that the open conformation is proteolytically inactive.

- Figure 5B: from the accompanying text, it is unclear where the data in Figure 5B comes from? The authors do not describe the DENV and WNV NS2B-3Pro purification and RNA binding assays. Please clarify. If the data was not generated in this manuscript, I don’t think it belongs here. Rather only the ZIKV NS2B-3pro IC50s. Perhaps the comparators could be placed in a table in the supplement, but it should be clear that this is not a direct comparison generated in parallel with ZIKV.

- Figure 6: The location of NSC83614 binding relative to the putative RNA binding channel the authors describe in Figure 6 should be described and discussed. Do the authors think that the compound binds in the RNA binding channel or promotes the super-open conformation to indirectly inhibit RNA binding in their model? Such discussion would help to better tie the interesting inhibitor data to the rest of the paper.

- Results: section entitled “Modeling RNA binding to ZIKV NS2B-3pro” – first sentence should read “… which raises an enticing possibility…”

- Figure 7: It is hard to understand why the authors chose to model this RNA with pseudoknot structures at either end, or why it is modeled in the figure. A ssRNA molecule as shown in Figure 6B is sufficient, and I don’t think the pseudoknots bring anything novel, rather they are just a source of confusion. I would remove this figure altogether for the sake of the story. At the very least the helical gate should be pointed out and an alternative view or insets should be shown to support the point that “[t]he modeling explains why RNA is processed from 3’-end toward 5’-end”.

- Figure 8-11: The authors should indicate the “helical gate” on the NS3 helicase domain. It is also unclear is Figure 8 is truly a main figure or belongs in the supplement. Fig 11 (model) shows the proposed “relevant conformation” and the flexibility point (regarding the linker) is not a novel idea and a minor point in my opinion.

- Figure 9 and 10 are missing from the manuscript, the latter has Figure 11 in front of it, making it impossible to evaluate the results.

- Figure 11: belongs in the discussion as it is not a new result, rather the output/proposed model of the results described herein.

- Figure 12: This figure doesn’t bring new information, rather is more of a discussion point and should be moved to the supplement.

- Potential Caveats and Limitations: belongs in the discussion not results section.

- Discussion: unclear what is meant by the double-negative, i.e. “…not be unable…” please clarify.

- Discussion: Which NS4 are you talking about here? NS4A, NS4B or both?

- One final point/issue I have as I return to this paper is that the authors state in the abstract, that they imagine that the contiguous binding surface binds to the “negative RNA strand exiting <the> helicase” as their data seems to support a model where ssRNA, but not dsRNA, can be bound by the NS2B-NS3 protease domain. However, I have trouble identifying when the single-stranded negative-strand would ever be accessible to the protease domain during infection. I.e. the NS3 helicase domain unwinds the dsRNA at the helical gate of the helicase, which routes the negative-strand through the helicase directly into the RdRp, where the positive-strand is synthesized from the negative-strand template. If this is true, the only time when the negative-strand is EVER single-stranded, would be when it is inside the helicase or RdRp. Thus, it is hard to imagine the scenario where the single-stranded negative-strand would be anywhere near or available for binding to the protease domain of NS3. It would always be in a dsRNA conformation when nearing the protease domain (i.e. during positive-strand RNA synthesis). Perhaps (based on the discussion section of the pepr), the authors are implying that the helical gate on the helicase domain is not in fact where the dsRNA is unwound/split and, that the protease domain is mediating the unwinding (contains some kind of helical gate itself) and only ssRNA is fed from the protease into the helicase domain? If this latter explanation is what the authors are proposing, perhaps they can point to the putative “helical gate” on the protease domain which would split the dsRNA in their model and discuss more about how the dsRNA would be proposed to be unwound here? As far as I can tell, the authors are simply proposing a contiguous ssRNA binding interface, not necessarily including a helical gate. Are the blue residues on either side of the ssRNA strand at the left-hand side of the protease actually thought to serve to “split” the dsRNA? This is VERY unclear from their putative model if so. Labeling of these residues would be helpful and pointing to the proposed residue that would serve as the helical gate would be helpful if so. Nonetheless, I think a more likely scenario is that during condensation of the viral RNA into the replication organelle, the positive-strand genomic RNA could come into contact with the protease domain of NS3 and this may be important for condensing the RNA into the replication organelle and/or synthesis of the negative-strand replicative intermediate. I think alternatively (although perhaps somewhat more speculative) it is also possible that the protease domain could bind to the viral RNA concurrently with translation and polyprotein synthesis, perhaps to aid the ribosome (or polyprotein folding) or as a mechanism to help retain the RNA for replication organelle biogenesis. I don’t think any of this necessarily negates the finding that the NS2B-3 protease can bind to ssRNA, just that unless they are indeed proposing that the protease is the domain where the dsRNA is unwound (and the helical gate is here), it is unlikely that it is the negative-strand RNA that is bound since the authors demonstrate that it does NOT bind to dsRNA. In the former scenario, the positive-strand is really the only viral RNA species which should be available to the protease domain in the context of the viral life cycle. The authors should thus adjust the text to reflect this, or provide additional discussion/justification for the scenario that would allow the negative-strand to be in a ssRNA conformation accessible to the NS2B-3 protease domain.</the>

PLOS authors have the option to publish the peer review history of their article (what does this mean?). If published, this will include your full peer review and any attached files.

Reviewer #1: No

Reviewer #2: No

Figure Files:

Data Requirements:

Reproducibility:

References:

---

## [Editor Report · Decision Letter 1]

2 Nov 2023

Dear Dr Terskikh,

We are pleased to inform you that your manuscript 'DUAL FUNCTION OF ZIKA VIRUS NS2B-NS3 PROTEASE' has been provisionally accepted for publication in PLOS Pathogens.

Best regards,

Sonja M. Best, Ph.D.

Section Editor

PLOS Pathogens

Sonja Best

Section Editor

PLOS Pathogens

Kasturi Haldar

Editor-in-Chief

PLOS Pathogens

orcid.org/0000-0001-5065-158X

Michael Malim

Editor-in-Chief

PLOS Pathogens

orcid.org/0000-0002-7699-2064
---

## [Editor Report · Acceptance letter]

17 Nov 2023

Dear Dr Terskikh,

We are delighted to inform you that your manuscript, "DUAL FUNCTION OF ZIKA VIRUS NS2B-NS3 PROTEASE," has been formally accepted for publication in PLOS Pathogens.

Best regards,

Kasturi Haldar

Editor-in-Chief

PLOS Pathogens

orcid.org/0000-0001-5065-158X

Michael Malim

Editor-in-Chief

PLOS Pathogens

orcid.org/0000-0002-7699-2064